**Subject Category:**
Biology (whole organism)

behaviour/evolution

sexual selection, fitness, personality, post-copulatory, sperm, temperament

**Author for correspondence:**
Clelia Gasparini
e-mail: clelia.gasparini@unipd.it

# The bold and the sperm: positive association between boldness and sperm number in the guppy

Clelia Gasparini[1,2], Elizabeth M. Speechley[1]
and Giovanni Polverino[1]

[1]Centre for Evolutionary Biology, School of Biological Sciences, University of Western Australia, Perth 6009, Western Australia, Australia
[2]Department of Biology, University of Padova, Padova, Italy

CG, 0000-0001-9172-1142; EMS, 0000-0003-0690-955X;
GP, 0000-0001-9737-7995

Assessing the consequences of personality traits on reproductive success is one of the most important challenges in personality studies and critical to understand the evolutionary implications of behavioural variability among animals. Personality traits are typically associated with mating acquisition in males, and, hence, linked to variation in their reproductive success. However, in most species, sexual selection continues after mating, and sperm traits (such as sperm number and quality) become very important in determining post-mating competitive success. Here, we investigate whether variation in personality traits is associated with variation in sperm traits using the guppy (*Poecilia reticulata*), a species with high levels of sperm competition. We found a positive association between boldness and sperm number but not sperm velocity, suggesting that bolder males have increased post-copulatory success than shyer individuals. No association was found between exploration and sperm traits. Our work highlights the importance of considering post-copulatory traits when investigating fitness consequences of personality traits, especially in species with high levels of female multiple matings and hence sperm competition.

## 1. Introduction

Boldness, aggressiveness and exploration are three behavioural traits often considered in studies on animal personality [1]. Despite the increasing interest in studying individual variability of behavioural traits in the last few decades, we still know little

about how variation in these traits is maintained within populations, and, in particular, the reproductive outcome of individuals differing in those traits. This is surprising as assessing the consequences of personality traits on reproductive success is crucial to understand the evolutionary implications of variability in those traits. Recent attention has therefore focused on the possible association between personality traits and reproductive success at the individual level. In males, personality traits are likely to be linked to mating success, mediated by a male's ability to defend a territory or resource, to win same-sex competition, to signal genetic quality and/or to find available mates [2,3]. Efforts so far have begun to unravel a complex interplay between a male's personality and his reproductive success. A meta-analysis across species revealed that being bold is linked to high reproductive success [4]. For example, bold bighorn sheep rams (*Ovis canadensis*) have been found to have a higher reproductive success than shy ones, with horn length being positively related to reproductive success [5]. In zebrafish, bolder males fertilize more eggs compared to their shyer counterparts [6]. However, there are some exceptions to this general trend such as in the hermit crab (*Pagurus bernhardus*), where less risk-prone males have increased reproductive success [7]. It would also be straightforward to think that a more explorative male would increase his chances to find an available mate, and hence his reproductive success, especially in low-density populations. In this direction, consistent among-individual variation in parental food provisioning and exploration is correlated in convict cichlid males (*Amatitlania nigrofasciata*) facing a predator threat [8]. However, studies so far have yielded mixed results on the association between exploration and reproductive success (e.g. [9]) and the meta-analysis did not support the hypothesis of a link between exploratory behaviour and reproductive success [4].

Importantly, a male's reproductive success is not only determined by his ability to obtain matings, but also by his ability to fertilize eggs [10]. This is especially true in species where females mate with multiple males during the same reproductive cycle (polyandry). To understand the link between behavioural traits and reproductive success, we therefore need to consider both pre- and post-mating episodes of sexual selection. Indeed, polyandry is ubiquitous in nature [11] and post-mating sexual selection is a strong evolutionary pressure that affects reproductive success in many species. When females mate with multiple males, a male's competitive fertilization success is affected by the number and the quality of his sperm [10], both positively linked to the outcome of sperm competition (where sperm from two or more males compete for the same set of eggs [12]). Interestingly, the link between behavioural traits and sperm traits has been mostly overlooked, and we still know very little on whether and to what extent behavioural traits vary with investment in sexually selected post-mating traits. The exception to this comes from studies on species with discrete (and fixed) male alternative tactics. In those species, males are usually either territorial or sneakers (satellites), and those reproductive tactics come with a specific suite of other morphological, behavioural and physiological traits, including different personality and sperm traits [13–15]. While in species with non-fixed alternative male mating tactics the association between boldness (or aggressiveness and exploration) and sperm traits (number and quality) is also expected [1,3], empirical support remains limited. It is particularly important to understand and disentangle the role of behavioural traits in both the stages of sexual selection (pre- and post-mating). Indeed, studies that so far have reported a link between behavioural traits and reproductive success (in terms of offspring sired) cannot disentangle whether the higher reproductive success is due to bolder males outcompeting rivals during mating acquisition or during sperm competition. For example, bolder and more aggressive zebrafish males were found to sire more eggs [6], but it was not possible to attribute the increased reproductive success to higher pre- or post-mating competitiveness.

Here, we fill this gap by studying the association between boldness, exploration and sperm traits (sperm number and quality) in the guppy (*Poecilia reticulata*). This species is ideal for our study as it is a well-known model species for sexual selection, and it has been the focus of many studies on both personality (e.g. [16–18]) and post-mating sexual selection (for a review see [19]). However, so far these two aspects have been studied independently, while here we look at both aspects within the same individual to build a bridge across this divide. We assessed boldness and exploratory behaviour (since aggressive behaviour is almost absent in this species [20,21]) and linked these traits to sperm number and quality. Importantly, by using the guppy, we can predict a male's post-mating success by assessing his sperm traits, as sperm number and sperm swimming velocity determine competitive fertilization success in this species [22]. Our main aim is to test whether those sperm traits are associated with an individual's willingness to explore and take risks in a novel environment.

# 2. Material and methods

## 2.1. Animals

Fish were maintained in standard laboratory conditions and reared in stock populations (1 : 1 sex ratio) at a temperature of 26 ± 1°C. A total of 60 males were used in the experiment, divided into four blocks for logistical reasons. We first measured sperm number and velocity and then we assessed the behaviour of each male three times, one week apart between each consecutive trial.

## 2.2. Sperm traits assessment

Sperm number and sperm velocity were assessed for each male before the behavioural assays. Each male was individually isolated in a plastic tank (2 l) for one week prior the sperm assays to standardize recent social and sexual experience. Each tank was physically separated from the others, but in visual contact with a tank containing females to keep the male sexually active and producing sperm (in this species if a male does not see females, his sperm production decreases [23,24]). To do so, we followed established protocols, whose details are reported in [25]. Briefly, each male was anaesthetized and sperm were collected after applying gentle pressure on the abdomen. Sperm were counted with a Neubauer haemocytometer and sperm velocity was assessed using a Hamilton-Thorne CASA system (sperm tracker). An average of 129 motile sperm was used in the sperm velocity analyses. After the sperm assays, each male was returned to its tank and the behavioural assays started one week after.

## 2.3. Boldness and exploratory assays

Boldness and exploration were tested according to standard protocols developed for guppies [17,18]. Each fish was first introduced inside a circular refuge (6 cm diameter) made of opaque PVC and placed in the centre of a circular arena (40 cm diameter). A video camera placed above the arena allowed us to observe the fish behaviour on a monitor, to avoid any disturbance. The fish was allowed to acclimatize inside the refuge for 5 min, and then the door was opened with a delicate rotating motion, to let him emerge at his own pace. We measured the time taken by an individual to emerge from a refuge and the amount of time the individual actively moved around a novel environment over 5 min (hereafter named boldness and exploration respectively). Short latencies to emerge from a refuge (emergence latency [26]) and extended time swimming around the novel environment (exploration [1]) indicate a willingness to take risks in exploring open spaces that are unfamiliar and potentially dangerous. After the test, each fish was gently netted from the arena and returned to its individual tank.

## 2.4. Statistical analyses

To test the association between personality (boldness and exploration) and sperm (number and quality) traits, we ran two separate linear mixed effect models (LMMs) in which boldness and exploration were included as the dependent variables. To help the interpretation of results, we inverted the scores for emergence time, so that higher scores of boldness indicate bolder individuals. In each LMM, block, trial (first, second or third), sperm number and sperm velocity were entered as fixed factors, while male ID was entered as a random factor (random intercepts) to account for repeated measurements. All LMMs were performed with the R package 'lme4' [27]. The adjusted repeatability [28] was estimated for each behavioural trait by calculating the proportion of the total phenotypic variance not attributable to fixed effects that were explained by among-individual variance. The significance of consistent among-individual differences (personality) was tested using likelihood ratio tests, whereby the full model including the individual as a random effect was compared with the reduced model in which the random effect was excluded. The significance of fixed effects was calculated from the F-statistic with the 'lmerTest' package [29] using Satterthwaite's approximation for the denominator degrees of freedom. Two individuals out of the 60 used did not complete the behavioural trials due to possible health issues, and therefore were excluded from the analyses. After the presence of personality was confirmed, we used the average scores of each individual for boldness and exploration to run a correlation between behavioural traits and sperm traits. All analyses were performed using R (v. 3.3.2) [30].

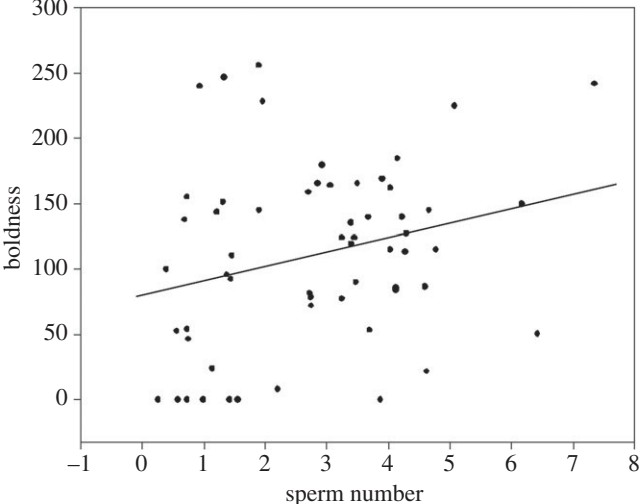

**Figure 1.** Correlation between boldness (average of three trials) and sperm number ($\times 10^6$).

**Table 1.** Results from the LMMs with boldness and exploration as dependent variables. Sperm number, sperm velocity, group and trial are included as fixed effects. Random intercepts are also included for each individual, which allowed variance decomposition. Intercepts ($V_{among}$), residuals ($V_{within}$) and adjusted repeatabilities are also shown for each behaviour. Significant $p$-values are indicated in bold type.

| fixed effects | estimate | $F$ | d.f.$_2$, d.f.$_1$ | $p$-value |
|---|---|---|---|---|
| boldness | | | | |
| sperm number | 12.015 | 4.275 | 1, 55 | **0.043** |
| sperm velocity | 0.462 | 0.848 | 1, 55 | 0.361 |
| group | −12.310 | 2.378 | 1, 55 | 0.129 |
| trial | 40.467 | 26.935 | 1, 117 | **<0.001** |
| random effects | variance | s.d. | $\Delta$AIC | $p$-value |
| $V_{among}$ | 2215 | 47.06 | 6.625 | **0.003** |
| $V_{within}$ | 7101 | 84.27 | | |
| repeatability | **0.215** | | | |
| fixed effects | estimate | $F$ | d.f.$_2$, d.f.$_1$ | $p$-value |
| exploration | | | | |
| sperm number | 10.283 | 3.375 | 1, 55 | 0.072 |
| sperm velocity | 0.591 | 1.489 | 1, 55 | 0.228 |
| group | 1.719 | 0.050 | 1, 55 | 0.824 |
| trial | −36.876 | 32.244 | 1, 117 | **<0.001** |
| random effects | variance | s.d. | $\Delta$AIC | $p$-value |
| $V_{among}$ | 2617 | 51.15 | 16.313 | **<0.001** |
| $V_{within}$ | 4923 | 70.16 | | |
| repeatability | **0.347** | | | |

## 3. Results

Boldness and exploration were both repeatable over time (see results from the LMMs in table 1). Sperm number, but not sperm velocity, explained a significant portion of the behavioural variance observed in boldness. Neither sperm number nor sperm velocity was associated with exploration (table 1). When using average individual scores for each behaviour, we found boldness to be positively correlated

with sperm number ($r = 0.26$, $n = 56$, $p = 0.046$, figure 1) but not with sperm velocity ($p = 0.759$), while no correlations were detected between sperm traits and exploratory behaviour (sperm number: $p = 0.13$, sperm velocity: $p = 0.55$) in agreement with the pattern of results found with the LMMs.

## 4. Discussion

Our findings revealed a positive association between boldness and sperm number, but not sperm velocity, in male guppies. On the contrary, exploration was not associated with sperm number or velocity. The pattern we found here is in line with the general pattern observed across species in the meta-analysis by Smith & Blumstein [4], where only certain personality traits correlated with reproductive success. Our study focuses on males and the link between their behaviour and sperm traits at the individual level. In the guppy, both sperm number and velocity are important in determining male post-mating success, with sperm number being the most important predictor of competitive fertilization success, and sperm velocity becoming important only when sperm number is equal among competing males [22]. The finding that boldness is positively associated with sperm number, therefore, is likely to translate in bolder individuals having a higher fertilization success at post-mating level than their shyer counterparts.

From an ecological point of view, the willingness of an individual to take risks and explore open spaces that are unfamiliar and potentially dangerous is the main target of selection [1] and it has been associated with many crucial ecological factors, including aggression, dispersal and invasiveness [4,31]. In the guppy, females have been shown to prefer bolder males, especially those males that during a simulated predator encounter were more likely to approach the predator [32], suggesting that boldness may be linked to increased reproductive success at pre-mating level. Importantly, boldness is likely to vary with predation pressure, with guppies from high predation sites being typically bolder than those from low predation localities [33]. A similar pattern of variation in boldness with predation pressure has also been observed in other poeciliids such as *Brachyraphis episcopi* [34]. Also, variation in predation levels has been suggested to modulate correlations between boldness and other fitness-related traits, such as growth, in *Gambusia holbrooki* [35]. As bolder individuals typically incur greater predation risks but increase their foraging opportunities and the number of females encountered, it is expected that a strong link between growth rates, sperm traits, reproductive success and personality traits will be found in male guppies at the individual level [3]. Similarly, the association between sperm number and boldness may vary between populations with different predation levels. Understanding the underlying mechanism linking boldness (referring here to a category of traits instead of specific behavioural traits, as suggested by [1]) and sperm number will be a fruitful avenue for future research, specifically focusing on genetic and hormonal factors, as well as environmental factors that may affect both the expression of boldness and spermatogenesis. Testosterone may be a promising starting point, as in many vertebrates it affects both behaviour and spermatogenesis [36].

Although significant, it is important to note that the among-individual variation in boldness explained by sperm number was low, so that the association we found between boldness and sperm number should be taken with caution at this stage until future studies will confirm this finding. Specifically, it will be interesting to see if the relationship between sperm number and boldness will become stronger when boldness is assessed in individuals from a different population from the one we used, and especially in wild individuals from a high predation site [33].

Our study adds a new aspect to the increasing body of work trying to understand the evolution, maintenance and fitness consequences of animal personality [1]. Boldness has been found to correlate with traits increasing pre-mating reproductive success, but traits linked to post-mating sexual selection have rarely been considered. It will be worth assessing whether personality and sperm traits are also associated in other species to fully understand the fitness consequences of personality, especially in species where the relative contribution of pre- and post-mating episodes of sexual selection is known. In conclusion, our findings highlight the importance of considering post-mating sexual selection when studying fitness consequences of personality, as pre-mating success (i.e. mate acquisition) alone may not tell the whole story.

Ethics. This research was conducted under the approval of the University of Western Australia's Animal Ethics Committee (approval no. RA/3/100/1376).

Data accessibility. Data are available from the Dryad Digital Repository: https://doi.org/10.5061/dryad.62f422p [37].

Authors' contributions. C.G. and G.P. conceived the study, conducted the experiment, analysed the data and wrote the manuscript. E.S. helped to collect behavioural data. All authors gave final approval for publication.

Competing interests. We declare we have no competing interests.

Funding. This research was funded by an ARC DECRA (DE150101625) to C.G. G.P. was supported by the Forrest Research Foundation.

Acknowledgements. We thank Jonathan Evans and Jens Krause for their useful comments and discussions during experimental design, and two anonymous reviewers for their useful comments on a previous version of the manuscript.

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
