## [Reviewer comments · Royal Society Open Science]

Review History

RSOS-190474.R0 (Original submission)

Review form: Reviewer 1 (Ingo Schlupp)

Is the manuscript scientifically sound in its present form?

Yes

Are the interpretations and conclusions justified by the results?

Yes

Is the language acceptable?

Yes

Is it clear how to access all supporting data?

Yes

Do you have any ethical concerns with this paper?

No

Have you any concerns about statistical analyses in this paper?

No

Recommendation?

Accept with minor revision (please list in comments)

Comments to the Author(s)

This is an interesting study that seeks to explore the connection between personality traits and sperm traits and establishes a link with post mating mate selection. The study finds a weak correlation between sperm number and boldness in males, but not exploration behavior. I suggest to replace the reported r with r -squared and to discuss that not much of the underlying variability is explained. Overall, however, the paper opens a door into a novel approach of investigating and understanding personality traits. I hope there is more to come!

Review form: Reviewer 2

Is the manuscript scientifically sound in its present form?

Yes

Are the interpretations and conclusions justified by the results?

Yes

Is the language acceptable?

Yes

Is it clear how to access all supporting data?

Not Applicable

Do you have any ethical concerns with this paper?

No

Have you any concerns about statistical analyses in this paper?

No

Recommendation?

Accept with minor revision (please list in comments)

Comments to the Author(s)

This manuscript addresses the literature gap found between personality traits and post-mating reproductive success, specifically they investigate the relationship between personality traits and different sperm characteristics. This manuscript was well written and to the point. However, there are some minor corrections I would suggest and questions I have about the study.

First, very simple to correct are minor grammar corrections:

Line 17: comma needed after "hence"

Line 30: comma needed after "aggressiveness"

Line 33: commas needed before and after "in particular"

Line 38: delete "a" found before "resource"

Line 40-42: In this sentence you mention that this often comes at the cost of increased mortality

and you follow this by giving 3 examples that have nothing to do with mortality but instead are examples of the previous sentence. Is line 40-42 necessary or is there a way to help clarify this up?

Line 68: comma needed after "behavioural"

Line 72-77: This sentence is hard to follow, I would suggest that you try to break it into two so that your reader isn't confused at the end of the paragraph.

Result section: I'm trying to understand why the p-values here don't match up with the ones in the table. You say it confirms your LMM results, but I'm not sure where they come from. Is the table from the LMMs and the ones reported in lines 148-149 from a regression? I'm confused, maybe it is me not completely following you statistical approach...

Line 162: commas needed around "therefore"

Discussion: Boldness is a very vague term, in your methods you define it as the time it takes a male to emerge from a refuge into a novel environment. But then throughout the discussion, you refer to boldness in males as his likeliness to approach predators or associated in some form with predation. Are these two definitions of boldness correlated and would the later definition also be correlated with sperm number? I think you are making an assumption here that may or may not be supported by your data. For instance, you may be "bold" when exploring a new environment, but you may not be bold when presented with an actual predator. Unless you can connect the two different definitions of boldness to one another, I think you should present this caveat in the discussion.

Decision letter (RSOS-190474.R0)

21-May-2019

Dear Dr Gasparini

On behalf of the Editors, I am pleased to inform you that your Manuscript RSOS-190474 entitled "The bold and the sperm: positive association between boldness and sperm number in the guppy" has been accepted for publication in Royal Society Open Science subject to minor revision in accordance with the referee suggestions. Please find the referees' comments at the end of this email.

The reviewers and handling editors have recommended publication, but also suggest some minor revisions to your manuscript. Therefore, I invite you to respond to the comments and revise your manuscript.

- Ethics statement

- Data accessibility

It is a condition of publication that all supporting data are made available either as supplementary information or preferably in a suitable permanent repository. The data accessibility section should state where the article's supporting data can be accessed. This section should also include details, where possible of where to access other relevant research materials such as statistical tools, protocols, software etc can be accessed. If the data has been deposited in an external repository this section should list the database, accession number and link to the DOI for all data from the article that has been made publicly available. Data sets that have been

deposited in an external repository and have a DOI should also be appropriately cited in the manuscript and included in the reference list.

If you wish to submit your supporting data or code to Dryad (<http://datadryad.org/>), or modify your current submission to dryad, please use the following link:
<http://datadryad.org/submit?journalID=RSOS&manu=RSOS-190474>

- **Competing interests**

- **Authors' contributions**

- **Acknowledgements**

- **Funding statement**

Because the schedule for publication is very tight, it is a condition of publication that you submit the revised version of your manuscript before 30-May-2019. Please note that the revision deadline will expire at 00.00am on this date. If you do not think you will be able to meet this date please let me know immediately.

When submitting your revised manuscript, you will be able to respond to the comments made by

the referees and upload a file "Response to Referees" in "Section 6 - File Upload". You can use this to document any changes you make to the original manuscript. In order to expedite the processing of the revised manuscript, please be as specific as possible in your response to the referees. We strongly recommend uploading two versions of your revised manuscript:

Kind regards,
Andrew Dunn

Royal Society Open Science Editorial Office
 Royal Society Open Science
 openscience@royalsociety.org

on behalf of Dr Michael Tobler (Associate Editor) and Kevin Padian (Subject Editor)
 openscience@royalsociety.org

Associate Editor Comments to Author (Dr Michael Tobler):

Associate Editor: 1

Comments to the Author:

We have received feedback from two reviewers, both of which very much liked the manuscript. Both have some minor comments that should help the authors improve the manuscript. I recommend that this manuscript is accepted for publication upon revision.

Reviewer comments to Author:

Reviewer: 1

Comments to the Author(s)

This is an interesting study that seeks to explore the connection between personality traits and sperm traits and establishes a link with post mating mate selection. The study finds a weak correlation between sperm number and boldness in males, but not exploration behavior. I suggest to replace the reported r with r -squared and to discuss that not much of the underlying variability is explained. Overall, however, the paper opens a door into a novel approach of investigating and understanding personality traits. I hope there is more to come!

Reviewer: 2

Comments to the Author(s)

This manuscript addresses the literature gap found between personality traits and post-mating reproductive success, specifically they investigate the relationship between personality traits and different sperm characteristics. This manuscript was well written and to the point. However, there are some minor corrections I would suggest and questions I have about the study.

First, very simple to correct are minor grammar corrections:

Line 17: comma needed after "hence"

Line 30: comma needed after "aggressiveness"

Line 33: commas needed before and after "in particular"

Line 38: delete "a" found before "resource"

Line 40-42: In this sentence you mention that this often comes at the cost of increased mortality and you follow this by giving 3 examples that have nothing to do with mortality but instead are examples of the previous sentence. Is line 40-42 necessary or is there a way to help clarify this up?

Line 68: comma needed after "behavioural"

Line 72-77: This sentence is hard to follow, I would suggest that you try to break it into two so that your reader isn't confused at the end of the paragraph.

Result section: I'm trying to understand why the p -values here don't match up with the ones in the table. You say it confirms your LMM results, but I'm not sure where they come from. Is the table from the LMMs and the ones reported in lines 148-149 from a regression? I'm confused, maybe it is me not completely following you statistical approach...

Line 162: commas needed around "therefore"

Discussion: Boldness is a very vague term, in your methods you define it as the time it takes a male to emerge from a refuge into a novel environment. But then throughout the discussion, you

refer to boldness in males as his likeliness to approach predators or associated in some form with predation. Are these two definitions of boldness correlated and would the later definition also be correlated with sperm number? I think you are making an assumption here that may or may not be supported by your data. For instance, you may be "bold" when exploring a new environment, but you may not be bold when presented with an actual predator. Unless you can connect the two different definitions of boldness to one another, I think you should present this caveat in the discussion.

Author's Response to Decision Letter for (RSOS-190474.R0)

See Appendix A.

Decision letter (RSOS-190474.R1)

04-Jun-2019

Dear Dr Gasparini,

I am pleased to inform you that your manuscript entitled "The bold and the sperm: positive association between boldness and sperm number in the guppy" is now accepted for publication in Royal Society Open Science.

Kind regards,
Alice Power
Royal Society Open Science
openscience@royalsociety.org

on behalf of Dr Michael Tobler (Associate Editor) and Kevin Padian (Subject Editor)
openscience@royalsociety.org

Appendix A

Dear Editors at RSOS,

We are grateful to you for considering our Manuscript (RSOS-190474) for publication in the Royal Society Open Science. We are delighted with the positive feedback from the Associate editor and both reviewers and grateful for their comments. We have now fully addressed these comments in our revised manuscript and in our detailed responses below.

Please do not hesitate to contact me if you require any further information

Yours faithfully,
Clelia Gasparini

Reviewer: 1

This is an interesting study that seeks to explore the connection between personality traits and sperm traits and establishes a link with post mating mate selection. The study finds a weak correlation between sperm number and boldness in males, but not exploration behavior. I suggest to replace the reported r with r -squared and to discuss that not much of the underlying variability is explained. Overall, however, the paper opens a door into a novel approach of investigating and understanding personality traits. I hope there is more to come!

R: We thank the reviewer for his/her positive feedback. We agree with him/her that, although significant, the among-individual variability in boldness explained by sperm number is relatively low, with the p -value from the correlation coefficient just below the significance threshold ($p=0.046$, Fig 1). We have discussed this aspect as suggested (see lines 188-193). Regarding replacing r with r -squared, we have preferred maintaining our original coefficient of correlation (r), also depicted in Figure 1, instead of the coefficient of determination (r -squared) that would have required a second, different, statistical approach.

Reviewer: 2

This manuscript addresses the literature gap found between personality traits and post-mating reproductive success, specifically they investigate the relationship between personality traits and different sperm characteristics. This manuscript was well written and to the point.

R: we are glad that also this reviewer found the paper interesting and appreciated its relevance in the field.

Line 17: comma needed after "hence"

R: done

Line 30: comma needed after "aggressiveness"

R: done

Line 33: commas needed before and after "in particular"

R: done

Line 38: delete "a" found before "resource"

R: done

Line 40-42: In this sentence you mention that this often comes at the cost of increased mortality and you follow this by giving 3 examples that have nothing to do with mortality but instead are examples of the previous sentence. Is line 40-42 necessary or is there a way to help clarify this up?

R: Correct. We deleted the part of the sentence that referred to mortality (i.e. "but often at the cost of increased mortality").

Line 68: comma needed after "behavioural"

R: Done

Line 72-77: This sentence is hard to follow, I would suggest that you try to break it into two so that your reader isn't confused at the end of the paragraph.

R: Thank you for pointing this out. We split and rephrase the sentence to make it easier to read.

Result section: I'm trying to understand why the p-values here don't match up with the ones in the table. You say it confirms your LMM results, but I'm not sure where they come from. Is the table from the LMMs and the ones reported in lines 148-149 from a regression? I'm confused, maybe it is me not completely following your statistical approach...

R: Table 1 reports the results (and P-values) obtained with the LMMs, while the P-values reported in the results section at L 152 and 153 are those from the correlations run using the average values for the four behavioural assays (details of this correlation are reported in the statistical analysis section at L 143-145). We have now specified what is reported in table 1 for clarity also at L 148.

Line 162: commas needed around "therefore"

R: Done

Discussion: Boldness is a very vague term, in your methods you define it as the time it takes a male to emerge from a refuge into a novel environment. But then throughout the discussion, you refer to boldness in males as his likeliness to approach predators or associated in some form with predation. Are these two definitions of boldness correlated and would the later definition also be correlated with sperm number? I think you are making an assumption here that may or may not be supported by your data. For instance, you may be "bold" when exploring a new environment, but you may not be bold when presented with an actual predator. Unless you can connect the two different definitions of boldness to one another, I think you should present this caveat in the discussion.

R: We thank the reviewer for the thoughtful comment. We note that the personality field has largely suffered from terminology issues until very recently, with the same behavioural trait called with different terms and different behavioural traits called with the same term. Recent work (see for example [1, 2]) have recommended using a simplified terminology for animal personality traits that is now widely adopted in the animal personality field. This approach relies on the 'Big Five' animal personality traits (i.e., boldness, exploration, activity, aggressiveness, and sociability) in which terms represent categories of traits instead of specific behavioural traits. Since our study overlaps the animal personality and the post-copulatory sexual selection research fields, we wanted to minimize potential issues related to the use of ambiguous terminology and, therefore, we have followed the conservative approach indicated by Réale et al. 2007 [2] and used the general category term "boldness" and "exploration" instead of more specific descriptive terms. Nevertheless, we agree with the reviewer that the terminology used here should account for different backgrounds of the audience and, therefore, we further clarified this in the text (see L 184-185 of the revised manuscript).

References

1. Carter A.J., Feeney W.E., Marshall H.H., Cowlshaw G., Heinsohn R. 2013 Animal personality: what are behavioural ecologists measuring? *Biol Rev* **88**(2), 465-475.
2. Réale D., Reader S.M., Sol D., McDougall P.T., Dingemanse N.J. 2007 Integrating animal temperament within ecology and evolution. *Biol Rev* **82**(2), 291-318.